# LOTIC: Long-Term Outcomes After Triple Arthrodesis in Children—A Retrospective Case Series

**DOI:** 10.3390/children13010029

**Published:** 2025-12-24

**Authors:** Angelina Arora, Tachelle Ting, Zoe Smith, Christy Graff

**Affiliations:** 1Department of Medicine, Flinders University, Bedford Park, SA 5042, Australia; 2School of Medicine, University of Adelaide, Adelaide, SA 5000, Australia; 3Department of Surgery, Women’s and Children’s Hospital, North Adelaide, SA 5006, Australia

**Keywords:** triple arthrodesis, children, long-term follow-up, patient-reported outcome measures

## Abstract

**Highlights:**

**What are the main findings?**
According to patient- and carer-reported outcome measures, over half of patients in this case series after paediatric triple arthrodesis still have pain in adult life.Satisfaction after surgery and improvement in functional outcomes in this complex patient cohort are also variable in adult life, with concerning rates of pain, lack of functional improvement and ongoing psychosocial impact, according to patients and their carers.

**What are the implications of the main findings?**
There was a significant risk in this cohort of patients having problems in adult life after having this surgery in childhood.Careful patient selection and pre-operative counselling, more modern pre-operative planning and surgical techniques, selective soft tissue balancing and longer-term follow-up may help future outcomes.

**Abstract:**

Background/Objectives: Triple arthrodesis (TA) involves fusion of subtalar, talonavicular and calcaneocuboid joints. In the paediatric population, this procedure is used to correct foot deformities, augment stability and decrease pain, often in neuromuscular conditions. There is limited research into long-term outcomes of paediatric TA in regards to patient quality of life (QOL). This study aims to retrospectively evaluate the long-term patient-reported outcomes of the paediatric TA at a single centre. Methods: All paediatric patients who underwent TA at the Women’s and Children’s Hospital between 1998 and 2012 were identified from operative records and the patient and/or their carer were given the opportunity to be interviewed. Patient-reported outcomes were measured over the phone using the Foot and Ankle Ability Measure (FAAM) and Manchester-Oxford Foot and Ankle Questionnaire (MOXFQ) tools. Results: Eighteen patients were included in the study, with a total of 23 feet, with five patients having bilateral surgery in the one sitting. Follow-up was at a mean time of 17 years post-surgery, with a mode of 20 years. There were recurring themes of continued pain plus impaired function and mobility, especially in children who could not walk prior to surgery. Derived from the FAAM, the average Activities of Daily Living Scale was 39.81%, with four patients at 0%. The MOXFQ outcomes for walking, pain and social interaction domains were converted from the Likert scale into an average total score of 34.99/64. Over half of the patients continued to have pain after the surgery in adult life. Conclusions: This study highlights variable results post-paediatric triple arthrodesis, with concerning rates of limited improvement in functional outcomes, ongoing pain and negative psychosocial impact in adulthood, as reported by the patient or their carer.

## 1. Introduction

Triple arthrodesis (TA) is the surgical fusion of the subtalar, talonavicular and calcaneocuboid joints. The procedure has historically been used as a definitive procedure to achieve a stable, plantigrade foot in children with complex hindfoot deformities. Originally described in the early 20th century as a salvage operation for paralytic or rigid foot deformities [1], the procedure remains an intervention for children with neuromuscular disorders such as cerebral palsy, spina bifida, Charcot–Marie–Tooth disease, and other conditions characterised by progressive deformity or instability of the hindfoot [2,3,4,5]. In these populations, progressive deformity may impair gait biomechanics, increase energy expenditure during ambulation, contribute to pain, prevent splinting and/or shoewear, and limit activities of daily living. Achieving durable correction is therefore considered essential for maintaining mobility and quality of life during adolescence and adulthood.

The goals of paediatric TA include improving foot alignment/shape, correcting bony deformity, reducing deformity recurrence, alleviating pain, improving foot function, enhancing brace tolerance and preventing ulceration or instability associated with neuromuscular imbalance [2,6]. Fusion of the triple joint complex also provides mechanical stability in children for whom soft-tissue balancing or joint-preserving procedures are insufficient or inappropriate. Short- and medium-term studies generally describe favourable radiographic correction and improvements in pain and function, particularly when the procedure achieves a plantigrade, braceable foot [2,6,7,8,9,10].

Despite its long history, evidence regarding long-term outcomes of paediatric TA is limited for children in later life [8,11]. The existing literature primarily focuses on surgeon-reported outcomes such as radiographic union, alignment, or complication rates as opposed to patient-reported quality of life [3,4,5,9,12]. Consequently, the true impact of paediatric TA on long-term quality of life, social participation and functional independence remains poorly understood.

There is also a recognised gap in understanding how early-life hindfoot fusion interacts with long-term neuromuscular disease progression, compensatory gait adaptations and degenerative changes in adjacent joints (such as the ankle or midfoot). These factors may contribute to functional decline or increasing pain in adulthood, yet few studies have tracked patients beyond skeletal maturity. Moreover, orthopaedic research lacks validated patient-reported outcome measures (PROMs) specifically designed for children or adults with disabilities, limiting the comparability and sensitivity of long-term follow-up data [13].

Given these evidence gaps, contemporary patient-reported outcomes following paediatric TA are not well understood. Understanding these outcomes is essential to guide family counselling, set realistic expectations and inform surgical decision-making for clinicians managing children with complex hindfoot deformities.

This retrospective study aims to address the following question: what is the long-term outcome for children who have triple arthrodesis on one or both feet in adult life, according to carer or patient reports?

## 2. Materials and Methods

Ethics approval was obtained through the Governance and Ethics Management System (GEMS), South Australia (2022/GEM00189) from the Women’s and Children’s Health Network South Australia Health Research Ethics Committee (WCHN HREC) on 6 October 2022. This single centre retrospective case series investigated paediatric patients (under 18 years old) who underwent triple arthrodesis at WCH from 1998 to 2012 and have now reached adulthood with a minimum of 10 years since their initial surgery. The period of 1998 to 2012 was chosen as 1998 was the year operative records were able to be identified using our current software, and 2012 was a minimum of 10 years prior to the year ethics approval was sought (2022).

Thirty participants were identified through operative records using the search terms “triple”, “arthrodesis”, “triple arthrodesis”, “subtalar arthrodesis”, “hindfoot fusion”, “triple fusion” and “hindfoot arthrodesis”. Patients that were deceased and those that did not speak English were excluded using demographic data obtained through Sunrise Electronic Medical Records and Oacis systems and the births, deaths and marriages register. Data were extracted from medical records including patient/carer details and details of the primary surgery including date, indication, complications and type of fixation.

A patient letter and information sheet regarding the study was sent to each participant. Informed consent was obtained from each patient or carer prior to the interview. Each patient and/or their carer was interviewed using a satisfaction score and the patient-reported outcome measures (PROMs) Foot and Ankle Ability Measure (FAAM) and Manchester Oxford Foot and Ankle Questionnaire (MOXFQ). The FAAM consists of an “Activities of Daily Living” subscale (21 scored items) in which the response options are presented as 5-point Likert scales (range 4 to 0). Scores for each subscale range from 0% (least function) to 100% (most function) (Martin et al., 2005) [14].

The Manchester Oxford Foot and Ankle Questionnaire (MOXFQ) is a validated 16-item patient-reported outcome measure for evaluating outcomes of foot and ankle surgery, answered on a 5-point Likert scale (each item is scored from 0 to 4, with 4 denoting most severe). In the MOXFQ, the questions are further sub-stratified into the categories of pain, walking and social impact. The satisfaction score was graded from very unsatisfied, somewhat unsatisfied, somewhat satisfied and very satisfied with an opportunity for reasoning.

De-identified data were recorded and stored on RedCAP 15.5.22 data software. Descriptive statistics only were applied due to the small sample size.

## 3. Results

### 3.1. Demographics

Thirty patients (with 41 feet undergoing triple arthrodesis, three patients in separate sittings and eight patients in the same sitting) were identified using the operative records search. Five patients were deceased. Eighteen patients (23 feet with triple arthrodesis, with five bilateral procedures in the same siting) were able to be contacted by phone and provided informed consent to proceed with the study (Figure 1). Two patients declined to participate, and five patients did not return our phone call after three attempts. The mean age of surgery was 14 years old and the mode age of surgery was 14 years old. There were 5 female patients and 13 male patients. The mean length of follow-up from the surgery was 17 years with a mode of 20 years after the date of surgery. Four patients could not walk prior to surgery. The indications for the triple arthrodesis procedure included a diverse range of conditions, as indicated in Table 1 below.

There were no significant differences between the included cohort and those not available for follow-up. In the cohort not available for follow-up, the mean age was 14.4 years at age of surgery, there were 10 males and 2 females, and the underlying diagnosis included 7 children with cerebral palsy, 2 children with tarsal coalitions, 2 children with chromosomal abnormalities and 1 child with an unknown condition.

### 3.2. Surgery

All patients had a single approach with bony resection to correct deformity. One patient had screw fixation; the rest had fixation with staples. Fourteen patients had bone graft from the bony resection placed back into the foot. Two patients had Achilles lengthening: one had an open Z lengthening and the other had a percutaneous Achilles lengthening. All patients were non-weightbearing in a cast for 6 weeks, then transitioned to weightbearing in a cast after this for 6 weeks, and then ankle foot orthoses (AFOs). All surgery was performed by consultant surgeons; there were six different surgeons performing the surgeries. The complications documented were return to theatre: two patients had an early return to theatre for a change of the cast due to swelling, one patient had removal of metalwork, and three patients later had surgery due to recurrence of the foot deformity. The details of these procedures were not available at the time of writing this paper, due to being undertaken at other sites.

### 3.3. Satisfaction

Seven of the eighteen patients were either “very unsatisfied” or “unsatisfied” on the satisfaction scale reflecting the outcome of the TA procedure in terms of their improvement of quality of life. This is shown in Table 2.

Reasons for dissatisfaction with the surgery are summarized in Table 3. Mobility status pre- and post-operatively are summarized in Table 4. Rates of patient satisfaction according to pre-operative walking status are summarized in Table 5.

### 3.4. FAAM and MOXFQ According to Pre-Operative and Post-Operative Walking Ability

The mean FAAM score was 39.8% (SD 33.1) across all 18 patients calculated from the Likert scores from their answers on the 21 items in the “Activities of Daily Living” subscale (Appendix A) (see Table 6). Four of eighteen patients had a total score of 0%, which means that they are unable to complete any of their Activities of Daily Living.

The mean MOXFQ score was 34.99 (SD 28.01) out of a total of 64 points calculated from the 16-item questionnaire relating to the amount, severity and impact of foot pain on function and activities of daily living. The higher the score, the more severe the pain and the worse the function (Appendix B) (see Table 6).

### 3.5. FAAM and MOXFQ Scores According to Underlying Diagnosis

The FAAM score is mainly related to function (Appendix A), and patients from each demographic scored an average of less than 50% (see Table 7). The MOXFQ score is related mainly to pain (Appendix B) and was higher for patients with conditions in which patients were mobile, such as club foot and tarsal coalition.

## 4. Discussion

The results of this study suggest that many patients after paediatric triple arthrodesis still have difficulty with pain and limited functional improvement a mean of 17 years after surgery, as reported by themselves or their carers. 67% of our population group had cerebral palsy or spina bifida. Fourteen patients could walk prior to surgery, and eleven could walk at final follow-up. 39% of our population group was either “somewhat unsatisfied” or “very unsatisfied” with their surgery results. 61% of the population still had pain in their foot/feet as an adult. Pain was worse in those with conditions that allowed more mobility, such as clubfoot and tarsal coalition. Surprisingly, the mean FAAM scores were similar in these subgroups, as compared to those less likely to be able to walk (such as cerebral palsy and spina bifida), suggesting their pain significantly impacted their function and activities of daily living. The mean scores for the modified PROMs were 39.8% (SD 33.1) for FAAM and 34.99 (SD 28.01) out of a total of 64 points for MOXFQ.

The results from our study correlate with the literature available. Salzman et al. (1999) [12] reported one of the largest series of paediatric triple arthrodesis into adult life, with 57 cases followed up at a mean of 44 years post-operation. A share of 55% of their population had ongoing pain at a mean of 44 years, and 78% had residual deformity. In this series, however, 91% said they would recommend the surgery, and 95% of patients were satisfied with the outcome, compared with our satisfaction rate which was 61% at a mean of 17 years. Reasons for this difference could include differences in soft tissue balancing and the population group. Two patients had a soft tissue balancing with Achilles lengthening in our series, compared with Salzman et al., in whom 48% of patients had a tendon transfer, and several patients staged procedures, with other surgeries including Achilles lengthening, plantar fascia release, midfoot fusion and midfoot osteotomy [12]. Over half the patients in their population had poliomyelitis, whereas most of our patient population had cerebral palsy or spina bifida. Wetmore et al. [5] had a “poor” result in 46.7% of their population group, an average of 21 years post-paediatric triple arthrodesis, all of whom had Charcot–Marie–Tooth (CMT) disease. They reported this was a worse outcome than in their poliomyelitis population and attributed this to the non-progressive nature of poliomyelitis [5]. They concluded that triple arthrodesis should only be performed in progressive neuromuscular deformities as a salvage procedure in fixed deformities, along with soft tissue balancing and tendon transfers. In contrast, Wukich et al. [4] had a good satisfaction rate of 86% at a mean of 12 years post-paediatric triple arthrodesis for CMT, although 68% of their population still had pain. Trehan et al. (2015) [3] and Tenuta et al. (1993) [10] reported dissatisfaction due to residual deformity and pain after this procedure, in keeping with our results. Vlachou et al. (2009) [9] reported 35 of 52 feet had a “fair” or “poor” result at an average of 10 years post-paediatric TA as determined by satisfaction questioning. They used a similar surgical technique of a single approach with bony resection, in which there is often significant soft tissue stripping and shortening of the foot. The literature review comparison is summarized in Table 8.

There are limitations of our study. We acknowledge the small sample size and loss to follow-up, which is largely due to the complex population and length of time after surgery. There were no significant differences between the included cohort and the cohort lost to follow-up. Our sample size is, however, comparable to that of other authors [3,5,9,10,15,16]. No radiographic data were included as the aim was to assess patient-/carer-reported outcome measures rather than radiographic or doctor-led clinical examination data. Ten of eighteen surveys were conducted with parents/carers of the patients as opposed to the patients themselves due to their disability. Carers know patients the best and have been observing and interacting with the patient the most over the years; they are the best proxy for the patient to report outcome measures. This is a complex patient cohort with significant morbidity, and the results of long-term follow-up also include progression of the patient’s disease state. We acknowledge the lack of pre-operative PROMs data, and therefore cannot truly quantify improvement or deterioration from pre-operative baseline. The FAAM score, being largely based on function, would have been significantly low pre-operatively in the diagnostic subgroups of severe cerebral palsy and spina bifida, and difficult to score both pre- and post-operatively. There are no validated PROMs for this patient population, including inter- or intra-rater reliability; the FAAM and MOXFQ are validated only for the verbal adult population. We recognize the need for validated PROMs in this complex population cohort [13], and more data and/or registries are needed to truly measure the long-term outcomes of this surgery. This is, however, one of the longest follow-ups for paediatric triple arthrodesis in the literature, with a mean of 17 years follow-up, and the only paper with PROMS follow-up. It gives valuable insights into outcomes of paediatric triple arthrodesis in this complex patient population.

More modern techniques of 3D computerized tomography (CT) planning now exist to better correct foot deformities with less bony resection. More modern implants and techniques for bone and soft tissue fixation, and less invasive surgeries with less soft tissue stripping, could potentially improve longer term outcomes for this surgery. Further research or long-term registries past childhood for paediatric patients undergoing this procedure are also needed to properly assess if these surgeries should be performed in childhood. From the results of this study and other studies in the literature, it is the opinion of this author that paediatric triple arthrodesis should be performed in children if

No other options exist to correct the deformity (such as osteotomy and/or tendon transfer) and/or severe deformities are present in children with conditions with unacceptably high relapse rates (such as non-ambulant cerebral palsy)Careful preoperative planning utilizing 3D CT is performed to improve foot shape and preserve foot length, as well as careful soft tissue balancing, follow-up past adulthood, and careful pre-operative counselling regarding the risk of residual and/or progressive deformity and longer-term symptoms.

## 5. Conclusions

This case series reports variable results in adult life after paediatric triple arthrodesis. It highlights concerning rates of patient difficulty a minimum of 10 years after paediatric triple arthrodesis, including limited improvement in functional outcomes, ongoing pain and negative psychosocial impact in the longer term, as reported by the patient or their carer. This adds to the current evidence in the literature for this patient cohort that patients may have pain and dissatisfaction in adult life after having this surgery in childhood. This descriptive retrospective case series aids to guide future prospective studies, to help improve patient- and carer-reported outcomes and satisfaction in the future for paediatric triple arthrodesis. Considerations may include careful patient selection and pre-operative counselling, accurate pre-operative planning, more modern techniques of surgical deformity correction and fixation, soft tissue balancing and follow-up beyond childhood.

## Figures and Tables

**Figure 1 children-13-00029-f001:**
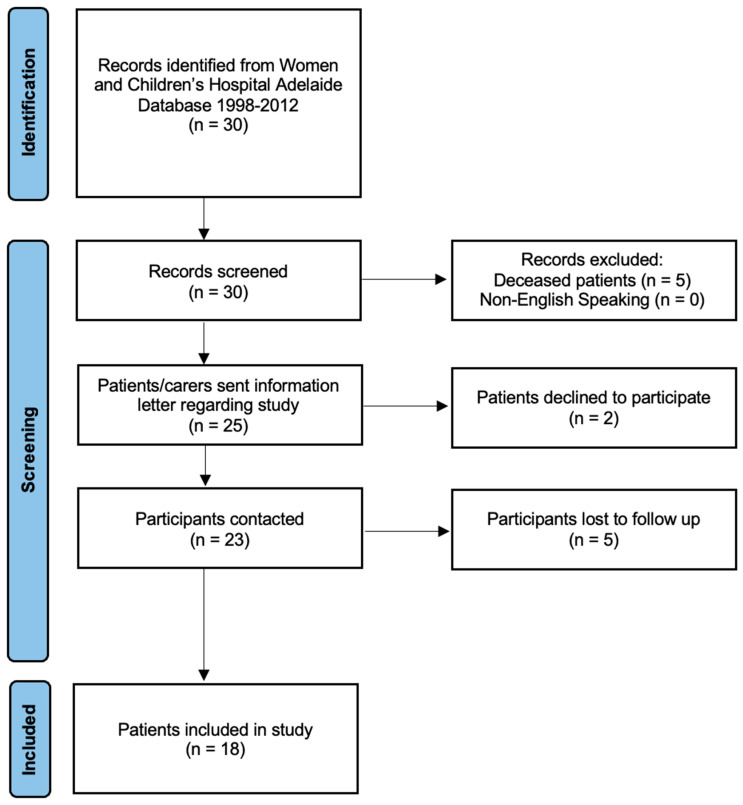
PRISMA Flow Diagram of final patient cohort.

**Table 1 children-13-00029-t001:** Patients included in study cohort separated by underlying diagnosis.

Underlying Diagnosis	Number of Patients
Cerebral Palsy	9
Spina Bifida	3
Club Foot	2
Tarsal Coalition	2
Post-Viral Foot Drop	1
Unknown/Other	2

**Table 2 children-13-00029-t002:** Satisfaction rating of patients in study.

Satisfaction	Number of Patients
Very Unsatisfied	1
Somewhat Unsatisfied	6
Somewhat Satisfied	6
Very Satisfied	5

**Table 3 children-13-00029-t003:** Patient/carer reasons for dissatisfaction after surgery.

Reason for Dissatisfaction	Number of Patients
Function not improved	15
Mobility not improved	14
Ongoing psychosocial impact	11
Ongoing pain	11
Unable to stand/walk	7
Shape of the foot not improved	5
Required further surgery	5

**Table 4 children-13-00029-t004:** Walking ability pre- and post-surgery.

Ability to Walk	Number of Patients	
	Preoperatively	Postoperatively
Able to walk	14	11
Unable to walk	4	7

**Table 5 children-13-00029-t005:** Satisfaction rating of patients according to their pre-op walking status.

Cohort	Satisfaction	Number of Patients
Walker	Very Unsatisfied	0
	Somewhat Unsatisfied	5
	Somewhat Satisfied	6
	Very Satisfied	3
Non-Walker	Very Unsatisfied	1
	Somewhat Unsatisfied	1
	Somewhat Satisfied	0
	Very Satisfied	2

**Table 6 children-13-00029-t006:** FAAM score of patients according to their operation and walking status.

Demographic	FAAM Score (%)	MOXFQ Score
All Patients	39.8	34.99
Pre-Op Non-Walker	16.3	43.62
Pre-Op Walker	46.5	28.65
Post-Op Non-Walker	10.5	31.58
Post-Op Walker	58.5	37.16

**Table 7 children-13-00029-t007:** FAAM and MOXFQ scores according to underlying diagnosis.

Condition	Number of Patients	Average FAAM	Average MOXFQ
Cerebral palsy	9	32.71	30.15
Spina bifida	3	41.6	23.67
Club foot	2	36.3	77.34
Tarsal coalition	1	46.3	53.12
Post-viral foot drop	1	0	16.66

**Table 8 children-13-00029-t008:** Key outcomes of relevant studies evaluating long term outcomes of TA in children.

Study	Key Outcomes
Trehan et al. [3]	10/26 had residual deformity with 38.1% continued pain
Tenuta et al. [10]	5/24 dissatisfied due to persistent pain and residual deformity 6/24 had ambulation limited by feet
Wetmore et al. [5]	14/30 poor result, 9/30 fair, 7/30 good result
Vlachou et al. [9]	53.8% fair result, 13.4% poor result
Wukich et al. [4]	15/22 complained of pain
Salzman et al. [12]	Results at a mean of 47 years: good 28%, fair 69%. 78% had residual deformity, 55% ongoing pain

## Data Availability

The original contributions presented in this study are included in the article. Further inquiries can be directed to the corresponding author.

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
