# Peer review of "LOTIC: Long-Term Outcomes After Triple Arthrodesis in Children—A Retrospective Case Series"

_children, 2025, doi:10.3390/children13010029_

Round 1
Reviewer 1 Report
Comments and Suggestions for Authors
The scientific work is well presented, and the data and results are clear and easily interpretable.
Among the positive aspects, the clarity with which the long-term outcomes are reported and the relevance of an extended follow-up stand out as additional strengths of this scientific project.
The limitations of the study are known and acknowledged by the authors, particularly its retrospective nature and the relatively small sample size.
With regard to potential improvements, I believe that:
-
It is necessary to expand the description of the surgical procedure performed (single versus double approach, fixation devices used, grafts or other required steps), as well as the number of surgeons involved and their respective levels of experience.
-
The statistical description of the sample (mean and mode) should be improved to provide greater clarity.
-
A detailed description of the postoperative protocol used should be included, specifying the follow-up schedule and the recommendations provided (for example, when weight-bearing was permitted or whether braces were used).
-
The complications observed should be described and analyzed in order to provide information regarding the most critical situations.
Author Response
- It is necessary to expand the description of the surgical procedure performed (single versus double approach, fixation devices used, grafts or other required steps), as well as the number of surgeons involved and their respective levels of experience.
Thank you for your comments. We have expanded this in the Results section.
- The statistical description of the sample (mean and mode) should be improved to provide greater clarity.
Thank you for your comment. We have described this in our results section.
- A detailed description of the postoperative protocol used should be included, specifying the follow-up schedule and the recommendations provided (for example, when weight-bearing was permitted or whether braces were used).
Thank you for your comment; we have added this to our results section
- The complications observed should be described and analyzed in order to provide information regarding the most critical situations.
Thank you for your comment. We have added this to our results section.

Reviewer 2 Report
Comments and Suggestions for Authors
General Impression
This study investigated patients with flatfoot who underwent triple arthrodesis, with a long-term follow-up period. Outcomes were assessed using patient satisfaction rates, FAAM, and MOXFQ scores. The primary concerns are the lack of discussion regarding the study’s results and inappropriate conclusions.
Abstract
- No specific comments.
Materials and Methods
- Lines 50–53: Please provide a description of the data extracted from the medical records.
- Line 54: A description of the satisfaction survey corresponding to Results section 3.1 is missing. Please add this information.
Results
- Lines 79–80 (Figure 1): The style of the flow diagram is unconventional. Please revise by adding side boxes to indicate the number of excluded patients and the reasons for exclusion.
- Lines 86–101: The presentation is somewhat confusing. Consider creating a table to make the information more concise and clearer.
- Line 102 (Table 3): The category “Very Unsatisfied” appears twice. Please check and correct.
- Lines 109–113: Please clarify that the maximum MOXFQ score is 64 points.
Discussion
- The discussion of the study’s results is insufficient. Please expand on this section. For example:
- Based on the findings, do the authors consider triple arthrodesis an acceptable treatment option for children?
- What types of patients do the authors believe are appropriate candidates for triple arthrodesis?
Conclusion
- Please ensure the conclusion is based on the results of this study. In addition, the conclusion in the main text should be consistent with that presented in the Abstract.
Author Response
This study investigated patients with flatfoot who underwent triple arthrodesis, with a long-term follow-up period. Outcomes were assessed using patient satisfaction rates, FAAM, and MOXFQ scores. The primary concerns are the lack of discussion regarding the study’s results and inappropriate conclusions.
Abstract
- No specific comments.
Materials and Methods
- Lines 50–53: Please provide a description of the data extracted from the medical records.
Thank you for comment. This has been addressed in the methods section.
- Line 54: A description of the satisfaction survey corresponding to Results section 3.1 is missing. Please add this information.
Thank you for your comment. This has been addressed in the methods section
Results
- Lines 79–80 (Figure 1): The style of the flow diagram is unconventional. Please revise by adding side boxes to indicate the number of excluded patients and the reasons for exclusion.
Thank you for your comment. The diagram has been revised
- Lines 86–101: The presentation is somewhat confusing. Consider creating a table to make the information more concise and clearer.
Thank you for the comment. The text has been summarised into 2 tables.
- Line 102 (Table 3): The category “Very Unsatisfied” appears twice. Please check and correct.
Thank you for your comments. This has been addressed.
- Lines 109–113: Please clarify that the maximum MOXFQ score is 64 points.
Thank you for your comment. This has been addressed.
Discussion
- The discussion of the study’s results is insufficient. Please expand on this section. For example:
- Based on the findings, do the authors consider triple arthrodesis an acceptable treatment option for children?
- What types of patients do the authors believe are appropriate candidates for triple arthrodesis?
Thank you for your comment. This has been addressed in the discussion
Conclusion
- Please ensure the conclusion is based on the results of this study. In addition, the conclusion in the main text should be consistent with that presented in the Abstract.
Thank you for your comment. This has been addressed throughout the paper and in the conclusion.

Round 2
Reviewer 2 Report
Comments and Suggestions for Authors
General Impression
The discussion and conclusion sections do not meet the standards required for publication and must be revised. The discussion is poorly organized, and the conclusion is vague, including statements not directly supported by the study’s results.
Abstract
- Line 36: Should this result be reported as 34.99/64 rather than 34.99/100?
Introduction
- No specific comments.
Materials and Methods
- Lines 86–114: Please divide this long paragraph into several shorter paragraphs for clarity.
Results
- Lines 116–141: Please provide a subsection title (e.g., “Demographics”) and renumber the subsequent subsections accordingly.
- Line 143: Please consider replacing the subsection title with a more appropriate one.
Discussion
- The discussion requires substantial revision to reach publication standards.
- Begin with a concise summary of the study’s results in the first paragraph.
- In subsequent paragraphs, compare the current findings with those reported in the literature and discuss them.
- State the study’s limitations in the final paragraph.
Specific points:
- Lines 176–179: This content should belong to the Introduction. Please delete it from the Discussion.
- Lines 180–181: This paragraph is meaningless and should be deleted. Avoid stating “this is the first paper regarding …,” as readers can understand novelty from the content itself.
- Line 183: The word “limitation” should not appear here. Delete it and instead introduce limitations at the beginning of the final paragraph of the Discussion.
- Lines 183–202: This section merely lists literature without discussion. Please compare your study results with those of other studies and address generalizability.
- Lines 203–216: Limitations should be described in the final paragraph of the Discussion.
- Lines 217–239: Please merge this section with lines 183–202 and restructure into new, topic-specific paragraphs that provide meaningful discussion.
Conclusions
- The conclusion is vague and unfocused. Please revise to ensure conclusions are based strictly on the study’s results.
- Delete any statements not supported by the findings.
- For example, the content in lines 242–245 is based on the study’s results but remains vague. Please make it more precise.
- The content in lines 245–251 is not directly based on the study’s results and should be removed.
- Provide succinct, results-driven conclusions that clearly state what was found and what can be concluded from this study.
Author Response
General Impression
The discussion and conclusion sections do not meet the standards required for publication and must be revised. The discussion is poorly organized, and the conclusion is vague, including statements not directly supported by the study’s results.
Abstract
- Line 36: Should this result be reported as 34.99/64 rather than 34.99/100?
Thank you for your comment. We have amended this.
Introduction
- No specific comments.
Materials and Methods
- Lines 86–114: Please divide this long paragraph into several shorter paragraphs for clarity.
Thank you for your comment. We have addressed this.
Results
- Lines 116–141: Please provide a subsection title (e.g., “Demographics”) and renumber the subsequent subsections accordingly.
Thank you for your comment. We have addressed this.
- Line 143: Please consider replacing the subsection title with a more appropriate one.
Thank you for your comment. We have addressed this.
Discussion
- The discussion requires substantial revision to reach publication standards.
- Begin with a concise summary of the study’s results in the first paragraph.
- In subsequent paragraphs, compare the current findings with those reported in the literature and discuss them.
- State the study’s limitations in the final paragraph.
Specific points:
- Lines 176–179: This content should belong to the Introduction. Please delete it from the Discussion.
Thank you for your comment. We have addressed this.
- Lines 180–181: This paragraph is meaningless and should be deleted. Avoid stating “this is the first paper regarding …,” as readers can understand novelty from the content itself.
Thank you for your comment. We have addressed this.
- Line 183: The word “limitation” should not appear here. Delete it and instead introduce limitations at the beginning of the final paragraph of the Discussion.
Thank you for your comment. We have deleted this.
- Lines 183–202: This section merely lists literature without discussion. Please compare your study results with those of other studies and address generalizability.
Thank you for your comment. We have amended this.
- Lines 203–216: Limitations should be described in the final paragraph of the Discussion.
Thank you for your comment. We have changed this.
- Lines 217–239: Please merge this section with lines 183–202 and restructure into new, topic-specific paragraphs that provide meaningful discussion.
Thank you for your comment. We have amended this.
Conclusions
- The conclusion is vague and unfocused. Please revise to ensure conclusions are based strictly on the study’s results.
- Delete any statements not supported by the findings.
- For example, the content in lines 242–245 is based on the study’s results but remains vague. Please make it more precise.
- The content in lines 245–251 is not directly based on the study’s results and should be removed.
- Provide succinct, results-driven conclusions that clearly state what was found and what can be concluded from this study.
Thank you for your comment. We have revised the conclusion.